# ALIGNING LARGE LANGUAGE MODEL BEHAVIOR WITH HUMAN CITATION PREFERENCES

## ABSTRACT

Most services built on powerful large-scale language models (LLMs) add citations to their output to enhance credibility. Recent research has paid increasing attention to the question of what reference documents to link to outputs. However, how LLMs recognize cite-worthiness and how this process should be controlled remains insufficiently explored. In this study, we focus on what kinds of content LLMs currently tend to cite and how well that behavior aligns with human preferences. We construct a dataset to characterize the relationship between human citation preferences and LLM behavior. Web-derived texts are categorized into eight citation-motivation types, and pairwise citation preferences are exhaustively evaluated across all type combinations to capture fine-grained contrasts. Our results show that humans most frequently seek citations for medical text, and stronger models display a similar tendency. We also find that current models are as much as 27% more likely than humans to add citations to text that is explicitly marked as needing citations on sources such as Wikipedia, and this overemphasis reduces alignment accuracy. Conversely, models systematically underselect numeric sentences (by $-22.6\%$ relative to humans) and sentences containing personal names (by $-20.1\%$), categories for which humans typically demand citations. Furthermore, experiments with fine-tuning and Direct Preference Optimization (DPO) demonstrate that model behavior can be calibrated to better match human citation preferences. We expect this study to provide a foundation for more fine-grained investigations into LLM citation preferences. Our dataset and code will be released upon publication.

## 1 INTRODUCTION

Large language models (LLMs) possess extensive knowledge about the world and hold the potential to fundamentally transform human society. In practice, powerful generative models such as the GPT series and Gemini are already deployed across a variety of downstream platforms, making the assurance of factuality and verifiability an important issue that affects many services. One technique for improving the verifiability of LLM outputs is the addition of citations. This means attaching, through some workflow, external documents that support the content generated by the LLM in the form of references. Most current high-performing closed models employ this functionality, and systems such as the GPT series(OpenAI, 2025), Gemini(Comanici et al., 2025), Claude(Anthropic, 2025), Qwen(Yang et al., 2025), and Perplexity[1] present citations to users.

The relationship between LLMs and citations has also drawn attention in recent research. In particular, the question of which documents should be linked to a given piece of text has been actively studied alongside RAG and AI-agent technologies, yielding many results. This line of work aligns well with the common LLM pipeline in which the system receives a user query, performs web search, and generates an answer based on information retrieved from the web (Google).

By contrast, the question of what information ought to receive a citation—that is, the importance of citations conditioned on the text content—has not been sufficiently investigated. Such research would enable citation behavior that aligns with user preferences and, crucially, would make it possible to control citation frequency. Too few citations undermine users' ability to verify claims and

---

[1] https://www.perplexity.ai

erode trust (Ding et al., 2025), whereas too many citations can reduce satisfaction, harm efficiency, and lower decision accuracy (De Jong, 2010; Eppler & Mengis, 2004), thereby degrading the user experience. In other words, a balance between verifiability and user experience is essential; rather than exhaustively presenting every source for a model's output, citations should focus on the most important information.

In this work, we address the open questions of whether LLMs align with users' citation preferences and whether models can be trained to do so. In today's high-performing LLM services, key decisions—such as which information to web-search and where in the output to attach citations—are often controlled by the LLM itself (Google; Anthropic). Thus, if we can teach LLMs users' citation preferences, we can adapt the citation behavior of AI-agent systems to match user needs.

To this end, we first conducted human annotation to investigate users' citation preferences. The data consisted of 6,000 Wikipedia sentences with quality labels, carefully annotated by Wikipedia editors. We grouped the quality labels into eight categories and, for each pair of categories, annotated—both with human preferences and with LLM preferences—which side should receive a citation (pairwise comparison). Our analysis showed that stronger LLMs exhibit higher alignment with human preferences. We also found that for sentences labeled "Citation needed" on Wikipedia, LLMs selected them at rates up to 19.5% higher for open models and up to 27.4% higher for closed models than humans did, indicating a strong influence from training data. Conversely, models systematically underselect sentences containing numbers (by up to $-22.6\%$ relative to humans) and sentences containing personal names (by up to $-20.1\%$), precisely the categories for which humans typically demand citations. Next, we trained LLMs using DPO and fine-tuning on the human-annotated data to align them with human preferences. As a result, we achieved an improvement of 11.8% and demonstrated that the influence of training data such as Wikipedia can be mitigated.

Our contributions are as follows.

- We present, to our knowledge, the first study that focuses on the need for citations within text and examines citation preferences of both humans and LLMs.

- We construct a dataset of 6,000 sentences with human citation-preference labels across eight content-based categories.

- Through dataset analysis, we show that LLMs are strongly influenced by their training data, which in part leads to divergences from human preferences.

- We demonstrate that DPO and fine-tuning can attenuate the impact of training data and bring model behavior closer to human citation preferences.

## 2 RELATED WORKS

Recent work has explored many ways of combining LLMs with citations (Lála et al., 2023; Gao et al., 2023). Examples include datasets that self-querying RAG methods in which the model decides at generation time whether external search is needed and, if so, retrieves and integrates paragraph-level evidence (Asai et al., 2024); tasks that require explicit source attribution for ambiguous QA (Shaier et al., 2024); and benchmarks that evaluate sentence-level citation in long-context QA (Zhang et al., 2024a).

There is also a line of work that evaluates the *validity* of citations (Li et al., 2024)—including studies that assess citation validity in the legal domain (Zhang et al., 2024b), datasets that test whether citations in generative search substantiate answers (Liu et al., 2023), and investigations of citation appropriateness in the medical domain (Wu et al., 2025). However, these efforts primarily focus on attaching an appropriate citation *to a given sentence*.

Closer to our work, there are a few studies that ask *which content should receive citations*. These include analyses of reasons for adding citations on Wikipedia (Redi et al., 2019) and careful examinations of citation intent in scientific papers (Wright & Augenstein, 2021; Saxena et al., 2024; Cohan et al., 2019). Such studies remain relatively scarce, and most target academic writing.

## 3 Task Definition

**Problem Setup**  Let $X$ be the set of sentences and let $K = \{1, \ldots, 8\}$ be the set of content categories. Each sentence $x \in X$ is assigned a category via a mapping $g : X \to K$.

A comparison item is a pair

$$p \in \{ (x_a, x_b) \mid g(x_a) \neq g(x_b) \}. \tag{1}$$

That is, the two sentences come from distinct categories in $K$. We balance the construction of pairs across unordered category pairs

$$\{k_a, k_b\} \subseteq K, \quad k_a \neq k_b, \tag{2}$$

so that each of the $\binom{8}{2} = 28$ category combinations is comparably represented.

## 4 Data Creation

### 4.1 Data Source

We require a collection of sentences annotated with quality labels as the data for our study. We adopt Wikipedia's Inline templates[2]—sentence-level tags that editors apply when some aspect of the content is problematic. These templates come in many varieties and are actively used across articles. Because inline templates tend to be used by relatively experienced editors, they serve as reasonably reliable signals.

In this study, we extract target sentences from the WikiSQE dataset curated from Wikipedia (Ando et al., 2024). This large-scale collection contains 3.4M sentences labeled with 153 categories. Although minimal noise filtering has already been applied, we further improve quality by manually identifying and removing broken sentences during annotation.

### 4.2 Category Group

We extract labels related to human citation preferences and reorganize them into eight categories. To this end, the authors reviewed all labels, selected 19 of them, and regrouped them into the following eight categories: *Missing Information*, *Sic*, *Doubt*, *Vague*, *POV*, *Medical Content*, *Jargon*, and *Unclear*. Details of the label–category mapping are shown in Table 1. *Missing Information* indicates that required details are absent; *Sic* flags typographical errors; *Doubt* marks statements of questionable veracity; *Vague* indicates imprecise or ambiguous wording; *POV* flags non-neutral or one-sided claims; *Medical Content* marks health/medical statements; *Jargon* flags highly technical or jargon-heavy wording; and *Unclear* indicates text that is difficult to understand.

### 4.3 Annotation

We annotated citation preferences using the collected dataset. We sampled 6,000 sentences to construct the annotation set: 750 sentences were drawn uniformly from each of the eight categories and paired to form 3,000 comparison items. Pairs were balanced across all $\binom{8}{2} = 28$ category combinations; that is, we created approximately 107 sentence pairs per category pair. During annotation, non-sentential items were flagged, two pairs per batch were duplicated for quality control, and only samples with fully consistent annotations were retained.

In total, 402 participants annotated the 3,000 pairs. All participants live in the U.S. We identified 404 pairs containing at least one non-sentence and removed them, yielding a final dataset of 2,596 pairs. Table 2 reports the human selection rates between categories.

*Medical content* sentences won broadly across categories, most notably against *Vague* (75.9%) and *Unclear* (66.3%), indicating a strong user preference to secure verifiability for medically consequential content. *Unclear* and *Jargon* were often favored or competitive, suggesting that citations are expected to serve as an "anchor of meaning" for hard-to-understand or highly technical sentences. Moreover, *Vague* exceeded *Missing Information* at 56.9% and *Unclear* exceeded *Missing*

---

[2]https://en.wikipedia.org/wiki/Category:Inline_templates

Table 1: Categories and brief descriptions of sentence labels for citation preference data, reorganized from Wikipedia inline templates.

| Category | Brief description |
|---|---|
| **Missing Information** | |
| Who? | Contains claims that do not identify individuals. |
| When? | Time period is so vague or ambiguous. |
| Which? | References to organizations or other things are vague. |
| Where? | Contains no specific place at which an event took place. |
| **Sic** | |
| Sic | Textual error in the statement is copied exactly from the source. |
| **Doubt** | |
| Dubious | Sourced statement, but that seems dubious or unlikely. |
| Disputed | Statement whose truth or factual is in dispute by editors. |
| **Vague** | |
| Vague | Contains vague words or statement. |
| Weasel words | Contains weasel words. |
| Ambiguous | Contains ambiguous phrases. |
| **POV** | |
| Neutrality disputed | Statement seemed to be biased. |
| Unbalanced opinion? | Statement may express a non-neutral point of view. |
| **Medical Content** | |
| Medical citation needed | Unsourced medical/health claim requiring citation. |
| **Jargon** | |
| Jargon | Overly jargonistic and too technical statement. |
| Expand acronym | Acronym/initialism should be expanded. |
| **Unclear** | |
| Clarification needed | Hard to understand; needs clarification. |
| Incomprehensible | Contains incomprehensible text. |

Table 2: Human preference win rates (%). Each cell is the share of judgments choosing the **row** category over the **column** category.

| Category | Info | Sic | Doubt | Vague | POV | Med | Jarg | Uncl |
|---|---|---|---|---|---|---|---|---|
| Info | – | 50.5 | 58.4 | 43.1 | 45.4 | 38.5 | 48.9 | 42.0 |
| Sic | 49.5 | – | 49.5 | 52.8 | 51.2 | 40.4 | 43.3 | 42.9 |
| Doubt | 41.6 | 50.5 | – | 48.9 | 56.0 | 38.5 | 53.5 | 51.7 |
| Vague | 56.9 | 47.2 | 51.1 | – | 42.2 | 24.1 | 43.8 | 41.2 |
| POV | 54.6 | 48.8 | 44.0 | 57.8 | – | 42.2 | 52.5 | 41.6 |
| Med | 61.5 | 59.6 | 61.5 | 75.9 | 57.8 | – | 57.3 | 66.3 |
| Jarg | 51.1 | 56.7 | 46.5 | 56.2 | 47.5 | 42.7 | – | 53.5 |
| Uncl | 58.0 | 57.1 | 48.3 | 58.8 | 58.4 | 33.7 | 46.5 | – |

*Information* at 58.0%, implying a tendency to prioritize readability and clarity with citations before filling in missing details.

## 5 CITATION PREFERENCE OF LLMS

### 5.1 SETUP

We evaluate LLMs using the dataset we constructed to study citation preferences. To cover a broad range, we include both open- and closed-source models as well as small and large models. Specifically, we consider 11 models: Mistral Small, Mistral Large, Llama 1B, Llama 3B, Llama 70B, DeepSeek Chat, GPT-5, Claude Sonnet 4, CommandR+, Gemini 2.5 Flash, and Qwen Max. For further details, see Appendix **??**. We attempted to collect outputs for all prompts; however, some models refused to answer certain items due to safety or political restrictions, resulting in up to three

Table 3: Agreement rates between models and humans by category (%). The agreement rate is the probability that a model's chosen sentence matches the human choice.

| Model | Info | Sic | Doubt | Vague | POV | Med | Jarg | Uncl | Avg |
|-------|------|------|-------|-------|------|------|------|------|------|
| Llama 1B | 50.2 | 48.7 | 49.2 | 50.4 | 49.1 | 50.9 | 52.6 | 48.7 | 50.0 |
| Llama 3B | 56.0 | 54.5 | 54.2 | 59.6 | 59.1 | 59.5 | 53.6 | 53.5 | 56.3 |
| Llama 70B | 61.9 | 61.2 | 61.0 | **64.4** | 58.7 | 65.6 | 59.8 | 60.3 | 61.6 |
| Mistral small | 56.2 | 57.8 | 54.6 | 55.8 | 57.8 | 60.0 | 55.2 | 60.8 | 57.3 |
| Mistral large | 55.8 | 61.2 | 61.7 | 58.1 | 58.3 | 61.4 | 59.8 | 60.7 | 59.6 |
| GPT-5 | 60.7 | 59.8 | 61.5 | 61.7 | 59.2 | 63.6 | 60.1 | 63.0 | 61.2 |
| Gemini | 56.8 | 58.9 | 58.3 | 59.2 | 59.5 | 65.4 | 57.6 | 61.3 | 59.6 |
| Claude | 59.8 | 61.0 | 62.0 | 62.7 | 59.6 | 64.7 | 61.3 | 61.3 | 61.5 |
| Deepseek | **64.0** | 61.1 | 61.3 | 61.6 | **61.8** | **67.1** | **61.9** | 62.6 | **62.7** |
| Qwen | 62.2 | **63.9** | **63.7** | 62.2 | 58.9 | 66.2 | 59.0 | **63.2** | 62.4 |
| CommandR+ | 55.9 | 54.4 | 58.9 | 56.8 | 55.4 | 59.1 | 55.6 | 56.1 | 56.5 |

missing samples per model. Moreover, open and closed LLMs may differ in how they handle citations. For open LLMs, everything hinges on the training data and training strategy, whereas closed LLMs may, in addition to the LLM itself, employ various tools or agent-like procedures. However, in our setting all closed LLMs are accessed via APIs that do not invoke any external components beyond the LLM itself. Therefore, this concern does not apply here.

## 5.2 ANALYSYS OF MODEL PREFERENCE

Table 3 presents the models' citation preferences. DeepSeek (62.7%) attains the highest overall performance, followed by Qwen (62.4%)), Llama-70B (61.6%)), Claude (61.5%)), and GPT-5 (61.2%)), revealing a clear parameter–scale effect. This pattern is especially pronounced within the Llama family; in particular, Llama 1B sits at the random baseline, implying that at this scale it effectively fails to produce meaningful citation preferences. Moreover, small LLMs are known to exhibit pronounced option–position biases in multiple-choice settings (Li & Gao, 2025), which likely contributes to this behavior.

Across categories, MEDICAL shows relatively high agreement with humans, suggesting that LLMs are comparatively adept at seeking citations for medical information. An interesting open question is at which stage of training this capability is acquired.

Nevertheless, aggregate scores plateau around 60%, indicating that current models only weakly predict human preferences and that the capability remains insufficient. These results suggest that inferring cite-worthiness implicitly from pretraining text alone is highly nontrivial—even for large, well-trained models—and remains a challenging task.

## 5.3 RELATIONSHIP BETWEEN INFORMATION AND CITATION PREFERENCES

We investigate what kinds of information drive the citation preferences of models versus humans from three perspectives.

**Citation needed** The first sentences containing "Citation needed." On Wikipedia, this is the label editors attach to mark that a citation is required, and it is among the most frequently applied inline templates. Because this label is common on Wikipedia, it is likely to appear frequently in LLM training data as well. If models are not explicitly de-biased with respect to citation behavior, we hypothesize that sentences bearing this label will be judged as requiring citations with high probability. (Note that "Medical citation needed" is a subtype of "Citation needed," and we include it in our analysis.)

The results are shown in Table 4a. With the exception of Llama 1B, all models select "Citation needed" substantially more often than humans. In particular, Llama 70B and DeepSeek exceed the human rate by more than +25%. This suggests a strong imprint of training-data biases that hinders alignment with human preferences. By contrast, models with similarly large parameter counts such as Mistral Large and CommandR+ stay within +5%, which hints that some corrective design in data

or training strategy may be at play. Given that Llama 1B is near the random baseline throughout our study, we do not discuss it further.

**Numeric sentences**  The second sentences containing numbers. Such sentences include dates and quantitative expressions, which demand high precision and leave little room for error. When precision is required, users are expected to rely on external sources to verify factuality. Indeed, prior work reports that users deem inline citations necessary for sentences containing statistics (Redi et al., 2019). Accordingly, we hypothesize that quantitative expressions increase users' demand for citations. We detect numeric sentences via simple pattern matching.

The results are shown in Table 4b. All models exhibit lower selection rates than humans. Notably, Mistral Small is 22.6% below the human rate. This indicates that models do not yet adequately capture users' citation preferences in this setting and that little specialized training for citation behavior has been conducted. Meanwhile, models such as GPT-5 and Claude show near-human alignment (within $-1.5\%$), again suggesting that some targeted design choices during training may contribute.

**Person names**  The third sentences containing personal names. When people are mentioned, particular care is required regarding factuality. Wikipedia explicitly flags this sensitivity [3], and biographies are tightly governed in practice. Because this caution is also widely recognized socially, we hypothesize that users similarly demand citations in such contexts.

The results are shown in Table 4c. As with "Numeric sentences," all models undershoot human selection rates. In particular, Llama 70B is 20.1% below the human rate. As before, this provides evidence that models insufficiently capture users' citation preferences. That said, Claude and CommandR+ align comparatively well with human behavior, consistent with the possibility of additional targeted design. A notable pattern is that most models fall below 50%, i.e., they often judge that a citation is not required in these cases.

## 6 ALIGNING LLMS WITH HUMAN CITATION PREFERENCES

Since the previous sections established that current LLMs are not aligned with users' citation preferences, we attempt to align them by training the models. If agreement with humans improves, this would constitute evidence that the models' citation-related behavior has been aligned. We consider five target models: Llama 1B, Llama 3B, Llama 70B, Mistral Small, and Mistral Large. For training and testing, we split the dataset constructed in Section 4 into two halves. The split is performed so that category pairs are balanced; when an odd count occurs, the extra items are assigned to the training set. In addition, we randomly sample 100 instances from the training portion as a validation set. As a result, the train/validation/test sets contain 1,206, 100, and 1,288 pairs, respectively. Further training details are provided in Appendix **??**.

### 6.1 FINE-TUNING

As a first approach, we perform fine-tuning. We use LoRA (Hu et al., 2022), which enables parameter-efficient training of LLMs with minimal loss in accuracy. For large models, we enable DeepSpeed ZeRO-Offload (Ren et al., 2021) to offload optimizer states and gradients to CPU memory.

The results are reported in Table 5. Overall, performance decreased. Consistent with prior findings that fine-tuning can change output format but does not reliably increase knowledge (Ghosh et al., 2024; Shengyu et al., 2023), it appears to have failed to inject knowledge about citation preferences. We conclude that standard fine-tuning is not well-suited for teaching citation preferences.

### 6.2 DPO

As a second approach, we apply DPO. As in fine-tuning, we use LoRA and DeepSpeed ZeRO-Offload. DPO is a commonly used method for preference optimization and appears to be the most suitable approach for our objective.

---

[3] https://en.wikipedia.org/wiki/Wikipedia:Biographies_of_living_persons

Table 4: Selection rates for citation-worthiness across evaluators and sentence types. "Rate" is the probability that the evaluator selected that sentence type; "vs Human" indicates the percentage change relative to the human selection rate. Bold numbers indicate the most pronounced values.

(a) Citation needed

| Evaluator | Rate (%) | vs Human |
|---|---|---|
| Human (Reference) | 58.7 | – |
| Llama 1B | 52.9 | -9.9% |
| Llama 3B | 66.1 | +12.6% |
| Llama 70B | 74.8 | **+27.4%** |
| Mistral small | 64.5 | +9.9% |
| Mistral large | 61.2 | +4.3% |
| GPT-5 | 68.6 | +16.9% |
| Gemini | 70.0 | +19.3% |
| Claude | 70.1 | +19.5% |
| Deepseek | 73.5 | +25.3% |
| Qwen | 68.6 | +16.9% |
| CommandR+ | 60.4 | +2.9% |

(b) Numeric sentences

| Evaluator | Rate (%) | vs Human |
|---|---|---|
| Human (Reference) | 61.9 | – |
| Llama 1B | 49.7 | -19.8% |
| Llama 3B | 55.3 | -10.8% |
| Llama 70B | 57.7 | -6.9% |
| Mistral small | 48.0 | **-22.6%** |
| Mistral large | 56.7 | -8.4% |
| GPT-5 | 61.6 | -0.6% |
| Gemini | 55.8 | -9.9% |
| Claude | 61.0 | -1.5% |
| Deepseek | 57.3 | -7.5% |
| Qwen | 59.6 | -3.9% |
| CommandR+ | 57.6 | -7.0% |

(c) Sentences with person names

| Evaluator | Rate (%) | vs Human |
|---|---|---|
| Human (Reference) | 51.7 | – |
| Llama 1B | 49.7 | -3.9% |
| Llama 3B | 45.2 | -12.7% |
| Llama 70B | 41.3 | **-20.1%** |
| Mistral small | 43.9 | -15.1% |
| Mistral large | 45.4 | -12.1% |
| GPT-5 | 48.5 | -6.2% |
| Gemini | 42.5 | -17.8% |
| Claude | 51.2 | -1.0% |
| Deepseek | 42.9 | -17.0% |
| Qwen | 47.4 | -8.3% |
| CommandR+ | 50.2 | -3.0% |

Table 5: Results after fine-tuning on the annotation data. Percent change relative to the non-fine-tuned baseline (%). We observe an average decrease in performance.

| Model | Info | Sic | Doubt | Vague | POV | Med | Jarg | Uncl | Avg |
|---|---|---|---|---|---|---|---|---|---|
| Llama 1B | -3.4 | 2.7 | 6.9 | -3.0 | 6.5 | -8.3 | -1.1 | 1.4 | 0.0 |
| Llama 3B | -14.5 | -10.1 | -8.7 | -19.1 | -15.7 | -21.0 | -10.8 | -18.5 | -14.4 |
| Llama 70B | -9.0 | -18.8 | -12.6 | -15.7 | -1.0 | -19.4 | -16.1 | -11.8 | -13.0 |
| Mistral small | 2.8 | -6.1 | 0.9 | 2.5 | 7.1 | -3.2 | 1.4 | -6.2 | -0.2 |
| Mistral large | 1.4 | -11.6 | -12.2 | -2.6 | -11.8 | -12.2 | -10.9 | -13.5 | -9.2 |

The results are shown in Table 6. Overall, performance increased. The gain is particularly pronounced for Llama 1B, whose agreement improves by 11.8% relative to its no-DPO counterpart. Llama 3B and Mistral Large also improve by roughly 9%, indicating that DPO is effective. By contrast, Mistral Small drops slightly, and Llama 70B declines by 1.6%. On average, we observe a 5.76% improvement, supporting DPO as a viable strategy for teaching models citation preferences.

Table 6: Results after applying DPO on the annotation data. Percent change relative to the non-fine-tuned baseline (%). We observe an average increase in performance.

| Model | Info | Sic | Doubt | Vague | POV | Med | Jarg | Uncl | Avg |
|---|---|---|---|---|---|---|---|---|---|
| Llama 1B | 3.2 | 8.6 | 19.3 | 14.7 | 6.9 | **15.9** | **12.5** | **14.0** | **11.8** |
| Llama 3B | 7.0 | **9.4** | **20.5** | 6.9 | 3.4 | 1.7 | 10.5 | 10.7 | 9.1 |
| Llama 70B | -5.2 | -2.6 | -3.1 | -6.1 | 2.0 | -5.3 | 4.5 | 2.0 | -1.6 |
| Mistral small | 2.8 | -6.1 | 0.9 | 2.5 | **7.1** | -3.2 | 1.4 | -6.2 | -0.2 |
| Mistral large | **12.4** | 8.2 | 7.9 | **16.2** | 2.7 | 9.9 | 11.9 | 9.1 | 9.7 |

## 7 CONCLUSION

We studied how LLMs decide *where* to add citations and how closely these decisions align with human preferences. Using 5,192 Wikipedia sentences reorganized into eight content-based categories and annotated via pairwise comparisons, we found that current models align with human citation preferences only weakly (low–60% agreement on average), with systematic divergences by sentence type. In particular, models substantially *overselect* sentences explicitly marked "Citation needed," while *underselecting* numeric and person-name sentences, both of which humans tend to treat as citation-worthy. Medical content emerges as the most consistently prioritized category across humans and models.

We then examined whether alignment can be improved through training. Standard fine-tuning generally reduced agreement, corroborating prior observations that conventional fine-tuning alters format but does not reliably instill new decision criteria. By contrast, Direct Preference Optimization (DPO) yielded consistent gains, with a mean improvement of 5.76% and especially large benefits for smaller models. These results indicate that preference-based training provides an effective mechanism for calibrating cite-worthiness judgments, whereas generic supervised fine-tuning is inadequate. Furthermore, our experiments demonstrate that LLM citation preferences can be controlled via DPO.

**Implications.** First, citation behavior should be *explicitly trained and evaluated* rather than assumed to emerge from pretraining. Second, deployment should consider *category-aware routing* (e.g., boosting numeric and person-name triggers) and *frequency control* to balance verifiability against information overload. Third, because many modern systems let the LLM implicitly govern search and attachment decisions, improving cite-worthiness alignment at the model level can directly benefit agentic pipelines.

**Limitations and future work.** Our analysis is bounded by Wikipedia-derived labels and English-only annotations; extending to other domains, languages, and high-risk classes (e.g., legal and scientific claims) is important. In addition, using larger-scale annotated datasets will enable a more rigorous examination of causality. Finally, integrating cite-worthiness prediction with retrieval/reranking and evaluating end-to-end user outcomes (task success, trust, cognitive load) remain open directions. To facilitate reproducibility, we plan to release the data and code upon acceptance of the paper.

## 8 ETHICS STATEMENT

The dataset we constructed includes opinions and descriptions originating from Wikipedia that may carry bias. Some items may contain offensive or discriminatory language. However, our objective in this study is to learn citation preferences *including* such real-world biases. We plan to release the dataset, with clear notices at distribution time that it may contain biased content.

We monitored annotators' working time and ensured that their pay always exceeded £6 per hour. We also paid annotators in full even if they failed the quality-control checks.

We used GPT-5 and Claude Code as AI tools; they assisted with searching and organizing prior work, checking text, and supporting code creation.

## 9 REPRODUCIBILITY STATEMENT

To facilitate reproducibility, we plan to release the data and code upon acceptance of the paper.

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

## A  ADDITIONAL DETAILS ON DATASET CREATION

In WikiSQE, the same sentence may appear under multiple labels. For pairwise construction we avoid pairing identical sentences with themselves; however, if duplicates occur across *different* categories, this is not problematic for our task, whereas duplicates *within* the same category group are filtered out. When assembling the data, we sample within each category so that counts are as balanced as possible across labels; if a label is underrepresented, we compensate by sampling from labels with larger pools. Although malformed sentences were discarded during annotation, the number of pairs per category after filtering is reported in Table 7. Finally, although LLMs are known to exhibit option–position bias (Li & Gao, 2025), in our setup the correct option is not fixed to a particular position, so this bias is not a concern.

Table 7: Head-to-head pair counts by citation category after filtering.

| Group | Info | Sic | Doubt | Vague | POV | Med | Jarg | Uncl | Total |
|-------|------|-----|-------|-------|-----|-----|------|------|-------|
| Info  | –    | 111 | 120   | 127   | 117 | 124 | 114  | 107  | **820** |
| Sic   | 111  | –   | 115   | 109   | 107 | 117 | 112  | 114  | **785** |
| Doubt | 120  | 115 | –     | 114   | 117 | 107 | 120  | 105  | **798** |
| Vague | 127  | 109 | 114   | –     | 96  | 113 | 112  | 115  | **786** |
| POV   | 117  | 107 | 117   | 96    | –   | 134 | 116  | 120  | **807** |
| Med   | 124  | 117 | 107   | 113   | 134 | –   | 111  | 115  | **821** |
| Jarg  | 114  | 112 | 120   | 112   | 116 | 111 | –    | 107  | **792** |
| Uncl  | 107  | 114 | 105   | 115   | 120 | 115 | 107  | –    | **783** |

## B EXPERIMENTAL SETUP

### B.1 MODELS

The models used for training are as follows. For GPT-5 we use the model snapshot as of September 1, 2025; for Claude we use `claude-sonnet-4-20250514`; for Gemini we use `gemini-2.5-flash` as of September 1, 2025; for Qwen we use `qwen-max-2025-01-25`; for DeepSeek we use `deepseek-chat` as of September 1, 2025; for CommandR+ we use `c4ai-command-r-plus-08-2024`; for Llama 70B we use `llama3-3-70b-instruct`; for Llama 3B we use `llama3-2-3b-instruct`; for Llama 1B we use `llama3-2-1b-instruct`; for Mistral Small we use `mistral-small-2402`; and for Mistral Large we use `mistral-large-2402`. To obtain high-quality responses, the decoding temperature is fixed at 1.0 for all models. During training, prompts are constructed following the official Mistral and Llama instruction templates.

