# OpenReview forum: "Aligning Large Language Model Behavior with Human Citation Preferences"
_ICLR.cc/2026/Conference — Submitted to ICLR 2026_

### Official Review · Reviewer_YstT · 2025-10-28

**Soundness:** 1
**Presentation:** 1
**Contribution:** 1
**Rating:** 2
**Confidence:** 5

**Summary:**

This paper studies how LLMs decide what to cite and how well this aligns with human preferences. The authors create a dataset categorizing texts into eight citation-motivation types and analyze pairwise citation choices. Results show LLMs over-cite obvious citation markers and under-cite numeric or personal-name sentences, diverging from human behavior. Further clues show that Fine-tuning and DPO can better align models with human citation preferences.

**Strengths:**

The authors create a dataset categorizing texts into eight citation-motivation types and analyze pairwise citation choices. Results show LLMs over-cite obvious citation markers and under-cite numeric or personal-name sentences, diverging from human behavior. The key strength of the paper is its systematic dataset and analysis, providing a detailed, fine-grained understanding of LLM citation behavior that can guide future improvements.

**Weaknesses:**

1. Missing details and unclear motivation: The paper lacks sufficient detail about the dataset construction process. It remains unclear why the authors chose to create this dataset, what specific research questions or gaps it aims to address, and how the base data sources were selected.

2. Uncertain data quality: The quality and reliability of the dataset are not properly evaluated. No assessment, validation, or inter-annotator agreement analysis is provided to ensure the soundness of the data.

3. Limited novelty: The paper does not present a new methodological contribution. It reads more like a technical report or dataset summary rather than a research paper with conceptual or algorithmic innovation.

4. Poor organization and writing issues: The motivation, contributions, and overall structure of the paper are unclear. The manuscript contains several typographical and formatting errors, which further reduce readability and weaken the presentation.

**Questions:**

1. It seems unusual that the authors recruited 402 participants to annotate only 3,000 sentence pairs. Such a large number of annotators could undermine the consistency of the dataset and introduce significant bias.

2. The problem setup in Section 3 is somewhat confusing. What is the underlying motivation for this task, and how exactly is it connected to citation preference?

3. The authors note that "some models refused to answer certain items due to safety or political restrictions." This raises ethical concerns about the dataset. Even though it is built from open-source data, it may still contain politically sensitive questions. Why were potentially unsafe items not removed?

4. Line 405 states, "Furthermore, our experiments demonstrate that LLM citation preferences can be controlled via DPO." This claim seems premature based solely on Table 6. Additional analysis is needed to support such a conclusion.

Typos:
Line 215, 305, broken links.

---

### Official Review · Reviewer_CqEW · 2025-10-31

**Soundness:** 2
**Presentation:** 2
**Contribution:** 3
**Rating:** 4
**Confidence:** 5

**Summary:**

This paper investigates how LLMs decide what content should receive citations and how well this aligns with human preferences. The authors construct a dataset of 6,000 Wikipedia sentences categorized into 8 types based on quality labels (Missing Information, Sic, Doubt, Vague, POV, Medical Content, Jargon, Unclear), with pairwise human annotations indicating which sentences in each pair should receive citations. They evaluate 11 LLMs (both open and closed) on this task and find that models only weakly align with human preferences (~60% agreement), with systematic biases: models overselect sentences explicitly marked "Citation needed" (up to +27.4%) while underselecting numeric sentences (-22.6%) and sentences with person names (-20.1%). The authors demonstrate that Direct Preference Optimization (DPO) can improve alignment by up to 11.8%, while standard fine-tuning fails. The work provides evidence that training data strongly influence citation behavior and that this influence can be controlled through preference-based training.

**Strengths:**

(A) This paper addresses a critical but underexplored aspect of LLM citation behavior - determining what content needs citations rather than which documents to link. This is particularly important because modern LLM services increasingly let the model itself control citation decisions in agentic workflows. The distinction between cite-worthiness (content-conditioned importance) and citation recommendation (document matching) is well-motivated and fills a genuine gap in the literature, as most prior work focuses on attribution and retrieval rather than the fundamental question of what merits verification.

(B) The dataset construction is thoughtful and rigorous. Using Wikipedia's inline templates provides a reliable signal from experienced editors, and the reorganization into 8 meaningful categories with balanced pairwise comparisons (28 category pairs) enables fine-grained analysis. The quality control measures (duplicate pairs for consistency checks, removal of malformed sentences) and annotation setup with 402 US-based participants is reasonable.

(C) he paper provides compelling evidence of how training data shapes citation behavior through three well-chosen analysis dimensions: (1) sentences with "Citation needed" tags show models are over-influenced by Wikipedia markup (+19.5% to +27.4% vs humans), (2) numeric sentences reveal systematic underselection (-22.6%) despite human preference for citing quantitative claims, and (3) person names show similar underselection (-20.1%) despite established norms around biographical verification.

(D) The experimental validation that DPO (but not standard fine-tuning) can improve alignment by 5.76% on average (11.8% for smaller models) provides actionable evidence that citation preferences can be learned and controlled. The negative results for fine-tuning corroborate prior findings about its limitations for knowledge injection, while the DPO success suggests a path forward for deployment.

**Weaknesses:**

(A) Minor: The final dataset contains only 2,596 pairwise comparisons, which becomes just 1,206 training pairs after splitting. This is quite small for training modern LLMs and limits the reliability of conclusions. More critically, the paper provides no inter-annotator agreement metrics (Cohen's kappa, Fleiss' kappa, or even pairwise agreement rates), making it impossible to assess whether the annotation task is well-defined and whether human preferences are consistent. The paper mentions removing inconsistent annotations but doesn't report what percentage of annotations were inconsistent, raising concerns about task difficulty and label reliability.

(B) The paper lacks essential methodological details that undermine reproducibility and credibility: (1) no prompts are shown for LLM evaluation, despite prompt sensitivity being well-known, (2) no statistical significance tests are reported - differences like 62.7% vs 61.2% could be within noise, (3) no confidence intervals or standard errors across evaluation runs, (4) no analysis of potential train/test contamination given that models were pretrained on Wikipedia and the dataset uses Wikipedia sentences.

(C)
The claim that models overselecting "Citation needed" sentences demonstrates "training data bias" (rather than appropriate learned behavior) assumes human annotations are ground truth without justification. However, Wikipedia editors explicitly marked these sentences as needing citations, so models selecting them more frequently could indicate they correctly learned Wikipedia's citation standards - which might be more rigorous than the crowdworkers' judgments. The paper doesn't validate that crowdworker preferences align with Wikipedia editor expertise or provide evidence that the model behavior is genuinely problematic rather than reflecting legitimate editorial standards that prioritize verifiability.

(D) The artificial balancing of category pairs (28 combinations with ~107 examples each) doesn't reflect natural distributions of content types requiring citations. Additionally, the eight categories seem somewhat arbitrary - the reorganization from 153 labels into 8 groups lacks clear principled justification, and some categories appear to overlap conceptually (e.g., "Vague" vs "Unclear," or why "Medical Content" is separate from domain-specific "Jargon").

**Questions:**

(A) What was the inter-annotator agreement rate (Cohen's kappa or Fleiss' kappa) for the pairwise comparisons? How many annotations were flagged as inconsistent and removed? Given that Table 2 shows many category pairs with win rates close to 50% (e.g., Info vs Sic: 50.5%, Doubt vs Vague: 48.9%), does this suggest the task may be ambiguous or poorly defined for certain category pairs? Could you provide evidence that crowdworkers' citation preferences align with those of experienced Wikipedia editors, given that Wikipedia's editorial standards may be more rigorous?

(B) Did you check whether the Wikipedia sentences in your dataset appear in the pretraining corpora of the evaluated models? This is particularly important since you argue models are influenced by seeing "Citation needed" tags during training - but this conclusion requires that models actually encountered these specific sentences or similar patterns. Additionally, could you provide the exact prompts used for LLM evaluation and report statistical significance tests (with confidence intervals) for the performance differences between models? Without these details, it's difficult to assess whether observed differences are meaningful or within experimental noise.

(C) our analysis shows models overselect "Citation needed" sentences (+19.5% to +27.4%) and underselect numeric/person-name sentences. However, couldn't the former indicate that models correctly learned Wikipedia's rigorous citation standards (which crowdworkers may underestimate), while the latter might reflect that numbers and names often appear in contexts where surrounding text provides attribution? Have you conducted ablation studies or controlled experiments to isolate whether these patterns are truly "biases" to correct, or whether they reflect reasonable learned heuristics? What evidence supports the assumption that crowdworker preferences should be preferred over model behavior that might better align with Wikipedia's actual editorial standards?

(D) How do you expect these findings to generalize beyond Wikipedia-style encyclopedic writing to other domains like scientific papers, journalism, or conversational AI responses where citation norms differ substantially? The category taxonomy seems Wikipedia-specific (e.g., "POV" reflects Wikipedia's neutrality policy) - have you considered how citation preferences might differ for medical literature (where citation standards are extremely rigorous), news articles (where attribution norms differ), or casual information-seeking (where users may prefer fewer citations for readability)? Could you discuss how the artificial balancing of category pairs in your dataset might affect model performance on natural distributions of content?

(E) The paper references missing appendices (e.g., "Appendix ??") multiple times for critical details about model training, hyperparameters, and DPO implementation. Could you provide complete details about: (1) DPO hyperparameters (learning rate, beta, number of epochs), (2) why standard fine-tuning failed so dramatically (Table 5 shows -14.4% for Llama 3B) - is this a hyperparameter issue or fundamental limitation, (3) how you constructed the preference pairs for DPO from pairwise human judgments, and (4) computational costs and training time? Additionally, given the small training set (1,206 pairs), did you observe overfitting, and how did you validate the models aren't simply memorizing the training data?

---

> ### Author Response · Authors · 2025-12-04
>
> Thank you very much for your detailed and wide-ranging comments.
> For the parts we are unable to fully address within the rebuttal period, we will conduct additional experiments and reflect them in the next revision of the paper.
>
> ## Regarding Question A
> The preference goal in our task is inherently subjective and thus fundamentally ambiguous to some extent. Precisely because of this, we believe that traditional inter-annotator agreement metrics are of limited interpretive value here: disagreement does not necessarily indicate label “noise,” but may simply reflect genuine variation in human preferences. In fact, many existing preference datasets for LLM alignment rely on signals that are not objectively defined and tolerate substantial subjectivity in judgments. Our view is aligned with this perspective: instead of forcing diverse user needs into a rigid, hand-designed notion of “correct” citation behavior, we see large-scale preference learning as the more appropriate way to capture and respect this diversity. Of course, we fully acknowledge that our current dataset is still small for this purpose, and increasing its scale is an important direction for future work.
>
> Concerning your concern that some category pairs may be inherently ambiguous: all sentences in our dataset are problematic in some way, so we expect that different annotators will sometimes disagree. For such pairs, a win rate close to 50% is, in our view, a natural reflection of the task difficulty rather than evidence of flawed category design.
>
> ## Regarding Question B
>
> The Wikipedia sentences we use are drawn from articles prior to 2022, so it is very likely that they were included in the pre-training data of many LLMs. However, because the exact training corpora of individual models are not fully disclosed, we cannot state this with certainty for each specific model.
> In addition, while there exist methods to test whether particular samples are included in a model’s training data, these techniques still do not allow us to make definitive, sentence-level inclusion claims for each LLM.
>
> ## Regarding Question C
>
> We do not have direct evidence that crowdworker preferences should be prioritized over Wikipedia editors’ standards. However, since the ultimate users of LLM-based services are end-users rather than Wikipedia editors, our view is that aligning citation behavior with ordinary users’ preferences is an important and legitimate design choice. In this work we therefore treat the crowdworkers’ judgments as a proxy for user-side preferences, while acknowledging that editor-level standards may be more rigorous in some respects.
>
> ## Regarding Question D
>
> In this work, we restricted ourselves to Wikipedia sentences with existing quality labels mainly for analytic convenience and controllability. We believe that the general framework can be extended to other domains by collecting analogous preference data there, and that such an extension would likely improve the external validity of our findings.
>
> We fully agree that domain-specific citation norms in specialized areas such as medicine, scientific writing, or journalism are highly interesting and important. However, we consider a thorough cross-domain investigation to be beyond the scope of the current paper and plan to treat it as future work.
>
> Regarding the artificially balanced sampling over category pairs, we acknowledge that the resulting distribution differs from the natural label distribution within Wikipedia. However, if the sampled sentences are reasonably representative of the underlying label groups, models that perform well under the imbalanced label distributions.

---

### Official Review · Reviewer_2zZU · 2025-11-01

**Soundness:** 3
**Presentation:** 2
**Contribution:** 2
**Rating:** 2
**Confidence:** 4

**Summary:**

This paper investigates what kind of content Large Language Models (LLMs) tend to cite and how well this behavior aligns with human preferences for "cite-worthiness." While many LLM services add citations to enhance credibility, this study explores a misalignment between LLM and human citation preferences.

The author created a new dataset by categorizing 6,000 Wikipedia sentences into eight "citation-motivation" types (e.g., Medical Content, Doubt, Vague, Unclear). They then conducted a large-scale study to capture human citation preferences by asking participants to choose which of two sentences from different categories most needed a citation.

At the end, they demonstrate a potential solution with DPO to train the models on human preference data resulted in an average 5.76% improvement in alignment, demonstrating that LLM citation behavior can be effectively calibrated to better match user needs.

**Strengths:**

- The paper clearly articulates the problem it aims to solve the gap between LLM citation behavior and human "cite-worthiness" preferences and outlines a straightforward methodology to address it
- The conclusions are supported. The paper evaluates 11 different models to identify the problem broadly, then trains and optimizes 5 open-source models to test its proposed solution
- A core contribution is the large, human-annotated preference dataset. The successful results from DPO (Direct Preference Optimization) training validate the feasibility of this approach, proving that model citation preferences can be effectively improved

**Weaknesses:**

- The writing of the paper need to be further crafted and polished. In line 214 and 205, there are missing appendix references. Redundancy is another issues exists, for example "fine-tuning hamrs alignment" are brought out repeatedly. I strongly suggest the author organise the logical flow and go through the paper carefully
- In the data collection phase, the description of the noise filtering process is insufficient. Additional information is required regarding "which noise filtering method was applied" and "how many instances/sentences remained at each stage of the filtering process."
- It is appreciable that the author has gathered a large group of participants for large-scale annotation work. However, this raises a concern regarding how to control the quality of human annotation, and the current paper may lack such a description
- The author categorizes the labels into eight categories, and this classification seems to lack justification for why the proposed definition is appropriate for decomposing human preferences

**Questions:**

Q1: Other than the three defined reference factors, what else could be essential for quantitatively assessing citation preferences?
Q2: What could be the underlying reason that the fine-tuning harm the citation alignment with human
Q3: When collecting the human annotation from such large group, how to make sure the human preferences are aligned within the group?

---

> ### Author Response · Authors · 2025-12-04
>
> Thank you very much for your helpful comments and for your suggestions on improving the writing. We will carefully revise the manuscript to reflect these points.
>
> Regarding **Weakness 2**, the Wikipedia sentences used in our study were collected and filtered in prior work, rather than being newly crawled by us. For detailed information about the original collection and filtering procedure, we kindly refer you to the data source paper cited in our manuscript.
>
> Regarding **Weakness 3**, we agree that the description of annotation quality control could be more detailed. In the revision, we will add a more thorough explanation of our quality-control pipeline, including how we filtered noisy annotations and how many instances remained at each stage.
>
> For **Question 1**, we agree that additional factors may be relevant for quantitatively assessing citation preferences. In this work, we focused on the three factors we could systematically operationalize. Beyond these, there are many potentially important aspects, such as lexical cues (e.g., the presence of hedges like *may* or *might* versus strong assertions), or whether a statement belongs to a high-stakes domain such as medicine, law, or finance. Exploring and incorporating such richer factors is future work.
>
> For **Question 3**, our view is that this task is inherently subjective, so we do not expect, nor necessarily desire, perfect alignment of preferences across annotators within the group. People come from diverse backgrounds, and their intuitions about which kinds of text “ought” to receive citations naturally differ. We believe that such subjectivity should be allowed, rather than forced into a single, artificially homogeneous notion of “ground truth.” disagreement does not necessarily indicate label “noise,” but may simply reflect genuine variation in human preferences. In fact, many existing preference datasets for LLM alignment rely on signals that are not objectively defined and tolerate substantial subjectivity in judgments. Our view is aligned with this perspective: instead of forcing diverse user needs into a rigid, hand-designed notion of “correct” citation behavior, we see large-scale preference learning as the more appropriate way to capture and respect this diversity. Of course, we fully acknowledge that our current dataset is still small for this purpose, and increasing its scale is an important direction for future work.

---

### Official Review · Reviewer_tDTx · 2025-11-01

**Soundness:** 2
**Presentation:** 3
**Contribution:** 3
**Rating:** 4
**Confidence:** 4

**Summary:**

The paper studies how large language models (LLMs) decide when to attach citations and how well those decisions line up with human expectations. Towards this goal, the authors obtained 6,000 sentences from Wikipedia that contain inline quality templates (e.g., “citation needed,” “clarification needed,” “medical citation needed”) and turned them into a preference dataset with human annotations. The authors evaluated most open-source large models on the curated preference dataset and used a train split to do SFT / DPO training.

**Strengths:**

1. The paper is centered “given two statements, which one most needs a citation?” which isolates cite-worthiness as a preference judgment. This is a novel task and different from most prior work on attribution, which tends to assume you already know a claim needs support and then focuses on finding or attaching the right source

2. The dataset curation is thoughtful and high quality. The use of Wikipedia inline templates is a very clever design, and this is accompanied by a large scale human annotation process with 400+ annotators. The result is a dataset that is both high volume and high quality, with interpretable labels.

**Weaknesses:**

My main concern is how useful this alignment goal itself is. The supervision signal is strictly relative (“which of two sentences needs a citation more?”). Real assistants, however, must make absolute, independent decisions about each span (“does this claim require a citation at all?”). Because the dataset never captures ‘both’, ‘neither’, or graded severity, it’s unclear whether models trained on this signal will learn a properly calibrated trigger for citation in deployment. Therefore even the human annotators may not often agree with each other (there's no inter-annotator agreement statistic), except for on clearly more important issues such as medical documents. There is a fundamental ambiguity with respect to the preference goal studied here.

The alignment experiments optimize directly on pairwise citation-preference agreement and then re-evaluate on that same objective. We don’t see any downstream, task-level validation (e.g., does the tuned model actually insert more citations for medically sensitive claims and fewer for benign filler, without just spamming citations everywhere?). Since the paper’s motivation is user trust and cognitive load in real assistant answers, an end-to-end generation study would be important to show that DPO improves practical citation behavior rather than overfitting to this pairwise benchmark.

**Questions:**

1. You describe performing annotator QC and filtering pairs for consistency before arriving at 2,596 comparisons. Could you report human–human agreement statistics on the unfiltered pool (e.g., raw pairwise agreement rate, κ / α), and also after filtering?

2. The best models achieve ~60% agreement with human preferences. How does this compare to (i) average agreement of a held-out group of annotators with the majority choice and (ii) majority-vs-majority agreement across annotator splits?

3. Can you share which of your eight categories have the most annotator disagreement? For example, are categories like “Vague / POV / Unclear” substantially noisier than “Medical Content”?

4. Can you show the performance of finetuned models on this preference dataset on some downstream citation tasks?

5. There are some frequently "preferred" topics such as medical claims. After DPO, do models become more likely to attach a citation to, e.g., medical claims? Do those citations actually support the claim any better?

---

> ### Author Response · Authors · 2025-12-04
>
> Thank you very much for your thoughtful comments and questions.
>
> ## Q1 (annotator agreement and QC):
> In this work, our annotator QC focuses on within-annotator consistency rather than between-annotator consistency. Concretely, within each batch we inserted duplicate pairs and retained only annotators whose judgments were self-consistent on these duplicates. Because of this design choice, we do not have overlapping labels across annotators for the final set of 2,596 comparisons, and therefore we cannot report human–human agreement statistics (e.g., κ/α) on the filtered pool.
>
> Related to your first weakness, we agree that the preference goal in our task is inherently subjective and thus fundamentally ambiguous to some extent. Precisely because of this, we believe that traditional inter-annotator agreement metrics are of limited interpretive value here: disagreement does not necessarily indicate label “noise,” but may simply reflect genuine variation in human preferences. In fact, many existing preference datasets for LLM alignment rely on signals that are not objectively defined and tolerate substantial subjectivity in judgments. Our view is aligned with this perspective: instead of forcing diverse user needs into a rigid, hand-designed notion of “correct” citation behavior, we see large-scale preference learning as the more appropriate way to capture and respect this diversity. Of course, we fully acknowledge that our current dataset is still small for this purpose, and increasing its scale is an important direction for future work.
>
> That said, we agree with you that additional duplicated annotations would be valuable—not only for evaluating models but also for characterizing which categories are intrinsically more ambiguous. As you suggest in Q2 and Q3, measuring majority-vs-majority agreement and per-category disagreement would yield useful insights. We plan to collect additional overlapping annotations in future iterations of the dataset and to include such analyses in a follow-up version.
>
> ## Q4 and Q5 (downstream citation behavior):
> We also agree that evaluating tuned models on downstream citation tasks would be highly informative. However, to the best of our knowledge there is currently almost no standardized benchmark that directly evaluates when an assistant should attach citations in free-form generation. Constructing such a dataset would require not only modeling generation, but also carefully annotating citation triggers and assessing whether retrieved citations genuinely support the underlying claims—both of which involve substantial additional data collection and evaluation design.
>
> Given these constraints, in this paper we chose to focus on the core preference benchmark and on demonstrating that DPO can improve alignment with human judgments on this benchmark. We see building robust downstream evaluation datasets for citation behavior (including medical claims, safety-sensitive content, and the quality of supporting evidence) as an important and challenging direction for future work rather than something we can adequately address within the scope of this submission. We will clarify this limitation and our future plans in the revised version.

---

### Official Review · Reviewer_6iHE · 2025-11-01

**Soundness:** 3
**Presentation:** 3
**Contribution:** 3
**Rating:** 4
**Confidence:** 3

**Summary:**

This paper studies alignment between LLM citation preferences and human preferences, constructing a dataset with 8 categories from Wikipedia. While the research direction is novel, the single-source dataset (Wikipedia only) \textbf{severely limits the generalizability and validity of conclusions.}

They found that current models diverge significantly from human preferences—models over-select sentences with [Citation needed] tags (27% higher than humans) but systematically under-select sentences with numbers and person names (22.6% and 20.1% lower, respectively).

Finally, they show that DPO training can improve alignment with human preferences.

**Strengths:**

**1. Novel and Timely Research Direction**
This paper focuses on an under-explored question that what content in LLM outputs deserves citations. Existing research centers on RAG retrieval, citation validation, or not cite-worthiness itself. Closest prior work (CiteWorth ACL 2021, Redi et al. 2019) is limited to narrow domains and excludes LLM behavior.

Its contributions fill this gap: (1) first use of preference learning for cite-worthiness; (2) cross-category comparison (8 types); (3) analysis of LLM-human alignment; (4) discovery of training data effects; (5) DPO demonstration for preference alignment.


**2. Well-Motivated Methodological Design**
The pairwise comparison framework captures relative preferences, avoiding absolute rating subjectivity. The 8-category taxonomy covers citation motivations, with balanced sampling across 28 pairs. Using Wikipedia templates is reasonable (due to editorial standards), though generalizability needs consideration.


**3. Comprehensive Model Evaluation Across Scales**
Evaluation includes 11 models (1B-70B+), spanning open-source (Llama, Mistral, DeepSeek) and closed-source (GPT-5, Claude, Gemini). Clear scale-performance correlation (Llama: 50.0%→56.3%→61.6%) shows how cite-worthiness capabilities emerge. Honest negative result reporting (Llama 1B at baseline) boosts credibility.


**4. Interesting Empirical Findings**
- Models over-select "Citation needed" sentences (up to +27.4%), revealing training data surface pattern influence.
- Consistent under-selection of numeric (-22.6%) and person name (-20.1%) sentences identifies systematic gaps.
- Medical content has higher agreement, suggesting domain-specific pretraining patterns.


**5. Practical Exploration of Alignment Methods**
A systematic comparison shows standard fine-tuning degrades performance, while DPO improves it by ~5.76%—aligning with recent preference optimization findings. Strong gains for small models (Llama 1B: +11.8%, 3B: +9.1%) add value for resource-constrained settings.


**6. Clear Presentation**
The paper is well-organized, with clear problem formulation and transparent reporting of results. Tables communicate findings effectively, and sufficient implementation details are provided. Committing to data/code release benefits future research.

**Weaknesses:**

The paper claims to study 'alignment between LLM and human citation preferences,' but its dataset has critical limitations that undermine this core goal:

1. **Single-domain bias**
All 6,000 sentences are sourced from Wikipedia, a specialized text type with unique editorial standards (e.g., prioritizing verifiability over practical utility). This differs fundamentally from ordinary users’ citation needs—for example, a user seeking 'insomnia medication advice' has distinct expectations vs. reading Wikipedia’s 'insomnia' entry. The study thus measures 'Wikipedia editor preferences' rather than general 'human preferences,' calling the validity of its research question into question.

2. **Circular reasoning in key findings**
A central claim (models over-select 'Citation needed' tags) is attributed to training data bias. However, annotation data is reorganized from Wikipedia inline templates (Table 1), creating tautology: Wikipedia labels define ground truth → models trained on Wikipedia are tested → models prefer Wikipedia labels → conclusion blames 'training bias.' This finding risks being a dataset artifact rather than a meaningful scientific discovery.

3. **Missing high-stakes real-world scenarios**
Critical application scenarios requiring citation control are absent, such as:
- High-stakes medical advice ('take XX medication') vs. medical knowledge statements ('XX drug mechanism is...')
- Legal opinions ('you have rights under XX law') vs. statute descriptions
Citation logic in these scenarios differs sharply from Wikipedia, yet the paper ignores them entirely.

4. **Inadequate sample size**
After cleaning, only 2,596 pairs remain (1,206 training, 100 validation, 1,288 test), with ~43 training pairs per category combination. This is too small to support learning 'general citation preferences' and instead risks model memorization of domain-specific patterns.

**Severity**: This is a fundamental flaw. The paper’s core claim ('LLMs misalign with human preferences') is compromised if it measures domain-specific preferences, relies on circular reasoning, and excludes critical scenarios. It reads more like a 'Wikipedia citation preference study' than a general analysis.

5. **Typo**
There are two '=Appendix ??' which are not correctly linked to the reference in the paper.

**Questions:**

**Question**
1. The study uses only Wikipedia data. Do you plan to add cross-domain datasets to test generalization? If not, how do you justify claiming the findings reflect 'human preferences' rather than Wikipedia-specific ones?
2. Concerning ablation studies: The paper does not test how prompts, training subset sizes, or sampling strategies affect results. Do you have unpublished ablation data to clarify these variables’ impact?


**Suggestions**

1. Add at least 2-3 datasets from other domains in humanities and social sciences, or conduct cross-domain validation demonstrating model performance across different domains. Maybe also increase the sample size accordingly.

If cross-domain data cannot be supplemented, I would suggest:
- Consider modifying the title to explicitly specify the research domain, such as 'Aligning LLM Behavior with Human Citation Preferences in Wikipedia Context' or similar.
- Clearly state in the Abstract and Introduction that this is a domain-specific study, and discuss generalization limitations in detail in the Conclusion and Limitations sections.
- Either remove the findings regarding 'Citation needed' (due to circular reasoning) or provide a reinterpretation of these results.

2. And, more ablation studies (different prompts, training data sizes, etc.) should be discussed.

---

> ### Author Response · Authors · 2025-12-04
>
> Thank you very much for your detailed comments. We believe there is one central misunderstanding we should first clarify, and then address your two questions.
>
> ## Question 1 (related to Weakness 1, 2, and 4)
>
> Your Weakness 1 (“Single-domain bias”) and Weakness 2 (“Circular reasoning in key findings”) appear to assume that our training signal directly reflects Wikipedia editor preferences. We apologize that our current wording did not make this distinction clearer.
>
> In our setup, the training data for preference learning are human annotations collected via pairwise comparisons, not Wikipedia template labels. The Wikipedia inline templates are only used to sample and categorize sentences into our eight “citation-motivation” types; the ground-truth preference labels used for both training and evaluation come from crowdworkers, not Wikipedia editors. We do not train on nor evaluate against the original Wikipedia template labels themselves.
> Because of this, we believe the strongest version of the “circular reasoning” concern does not apply to our method:
>
> - We do use Wikipedia as a text source,
> - but we do not use Wikipedia’s own labels as the supervisory signal for model training or evaluation.
>
> We fully agree that our dataset is domain-specific to Wikipedia-style encyclopedic text, and we will clarify this limitation more explicitly in the title/abstract/introduction and in the Limitations section. However, we do not think our setup reduces to purely measuring “Wikipedia editor preferences,” nor that the core findings rely on a tautological loop between Wikipedia labels and model training.
>
> We understand that Weakness 4 (“this reads more like a Wikipedia citation preference study than a general analysis”) likely stems from this misunderstanding.
>
>
> ## Regarding Weakness 3 (missing high-stakes real-world scenarios)
>
> We appreciate your concrete suggestions (e.g., medical advice vs. knowledge statements, legal advice vs. statute descriptions). We agree these domain-tailored, high-stakes scenarios are very important; however, designing and evaluating such domain-specific, end-to-end assistants is beyond the scope of this initial study. We see our work as a foundational step that isolates cite-worthiness as a preference signal; extending this to high-stakes domains is a natural and valuable direction for future work, and we will emphasize this more clearly in the revised paper.
>
>
> ## Question 2 (ablations on training size, prompts, and sampling)
>
> We agree that more detailed ablations on training set size, sampling strategy, and prompt variants would further strengthen the paper, and we appreciate this suggestion.
> However, we clarify that we view our current experiments as establishing stable qualitative patterns and a proof of concept for preference-based alignment of citation behavior, rather than an exhaustive exploration of all training configurations.
> Explicitly discuss the potential impact of prompt choice, training size, and sampling strategy in the Limitations section.

---

### Meta-Review · Area_Chair_Gozy · 2026-01-07

**Summary:**

This paper studies the gap or the extent of behavior alignment between LLM citation preference and human citation preference. To analyze this gap, the authors construct a dataset containing 6000 sentences from wikipedia and categorize them into eight citation-motivation types. Experiment result show that despite a similar tendency toward medical text, models are more likely to add citations that are marked citation needed than humans, while less likely to add citations on texts with numerical values and person names where humans would prefer. Lastly, the authors propose to use DPO to align human preference with the language model.

Overall, this paper raises a very interesting question that is under-investigated: how LLMs prefer citations compared with humans. The 8-category taxonomy and pairwise evaluation are comprehensive and rigorous. Experiments are extensive, ranging over a wide list of open-source and closed-source models with varying scales. The findings on citation preference gap between LLMs and humans are interesting and informative, offering new insights toward future preference alignment research.

Beyond those, the reviewers point out the following weaknesses that deserve further investigation and revision:
- The study only uses wikipedia as the data source for evaluation. This limits its applicability to other domains and questions whether the findings are sound beyond wikipedia domain.
- The validity of this measurement is questionable. The pairwise evaluation is subjective, and the authors did not consider using multiple annotators per sample. It is not known how ambiguous or subjective the human annotations are. This makes the problem setting less rigorous.
- Implications of DPO with alignment data on realistic applications are missing. The experiments only validate training with DPO on the same pairwise preference task. It is not known how this training changes the citation behavior of LLM in real tasks.
- How to control the quality of human annotation is missing.
- The results lack significance tests and the size of the training data is small.
- Missing details of the method. The presentation needs to be refined.

**Reviewer Concerns:**

Concerns being addressed:
- Clarification of problem setting and the implication of the different preferences from LLMs and humans.

Outstanding concerns:
- The study only uses wikipedia as the data source for evaluation. This limits its applicability to other domains and questions whether the findings are sound beyond wikipedia domain.
- The validity of this measurement is questionable. The pairwise evaluation is subjective, and the authors did not consider using multiple annotators per sample. It is not known how ambiguous or subjective the human annotations are. This makes the problem setting less rigorous.
- Implications of DPO with alignment data on realistic applications are missing. The experiments only validate training with DPO on the same pairwise preference task. It is not known how this training changes the citation behavior of LLM in real tasks.
- How to control the quality of human annotation is missing.
- The results lack significance tests and the size of the training data is small.
- Missing details of the method. The presentation needs to be refined.

**Reviewer Scores:**

I don't think the reviewer scores would change.

---

### Decision · Program_Chairs · 2026-01-26

Reject